# Practical Application of Plan–Do–Check–Act Cycle for Quality Improvement of Sustainable Packaging: A Case Study

**Vi Nguyen [1], Nam Nguyen [1], Bastian Schumacher [2] and Thanh Tran [1,*]**

[1] Faculty of Engineering, Vietnamese-German University, Le Lai Street, Hoa Phu Ward, Thu Dau Mot City 75000, Binh Duong Province, Vietnam; vi.nh@vgu.edu.vn (V.N.); gpem2016_nam.nh@student.vgu.edu.vn (N.N.)

[2] Chair of Sustainable Corporate Development, Technische Universität Berlin, Straße des 17. Juni 135, 10623 Berlin, Germany; bastian.schumacher@tu-berlin.de

[*] Correspondence: thanh.tt@vgu.edu.vn



**Featured Application: This research underscores the benefits of Plan–Do–Check–Act (PDCA) methodology in quality improvement. Practitioners can use it as the simplified guidance to practice PDCA combined with other support tools. Through the packaging case study, all stages of PDCA are clearly instructed step by step to effectively implement. The new packaging method which concentrates on effective designs and using recycled, bio-degradable, friendly environmental materials balances quality and profit for the company. It can be used as a benchmark example for PDCA in continuous quality improvement for packaging.**

**Abstract:** The research aims to give practical instructions for applying Plan–Do–Check–Act (PDCA) cycle in a packaging process. Eco-friendly, recycled material and a new packaging method for quality improvement and cost efficiency of heavily fragile product packaging are studied in this paper. A case study was conducted at GPEM laboratory, Vietnamese German University, Vietnam. In this case study, the current packaging style with Styrofoam material was analyzed and replaced by new packaging material and methods after applying the PDCA cycle for continuous quality improvement. Targets of the research were to find the new packaging method using friendly environment materials, to improve the quality, and to reduce the defect ratio due to packaging for fine-stone round surface fountains. Moreover, the extra cost should not be higher than 20% compared with the current packaging cost. The article proposes a simplified way that focuses on the combination of quality tools in the PDCA multiple phases to solve these problems. The quality tools are applied effectively through the PDCA cycle from collecting data, defining, analysis, testing, evaluation, and making decisions. New packaging design was been produced and tested successfully. One hundred percent of new packaging boxes for the mid-weight fountains (under 15 kg) passed the dropping test condition. Nearly 10% of the heavier weight products (above 15 kg) still had some small cracks on their top and bottom due to drop tests. Another PDCA cycle is recommended to continue applying for achieving a thorough solution. The conducted results show that PDCA is an effective method to tackle the damage product issue due to inappropriate packaging material and technique. It also brings good solutions for balancing sustainable packaging improvement and reducing the cost to ensure profit for companies. Besides contributing a guide reference for PDCA deployment, the authors intend to inspire practitioners and researchers to broaden exploration of the PDCA applications for sustainable packaging methodology. The research analysis shows that the PDCA methodology should be applied for defect reduction and quality enhancement in the packaging field. The field currently lacks systematic guidance for continuous improvement.

**Keywords:** lean manufacturing; PDCA; sustainable packaging; quality management

## 1. Introduction

Packaging has an important role not only in increasing the customer's satisfaction with the product but also in improving the economic profits of a company. More than the function of containing products inside, good packaging protects products from damages in transport, storage, and logistics; from the end of a production line delivered over retailers, till the end customers. Also, the packaging is an effective mean of conveying information to customers and partners through design labels shown by product color, manufacturing date, ingredients, characteristics, weight, user guide, etc. These results in reducing warranty and compensation costs, increasing the customer's satisfaction and company' reputation. In recent years, developing and using sustainable packaging is a global concern to protect the environment and to reduce the burden of waste treatment for the future generation. Trends in packaging industry development concentrate in a lower cost: Saving energy on packaging processing; reducing raw materials; using recycled, bio-degradable material; making a package with prolonging the durability and flexible for reuse; developing smart and interactive packaging to consumers [1]. This trend also poses challenges for companies in finding new materials and packaging methods which are able to enhance their responsibility in protecting the living environment and community but still ensure their economic returns.

There are many types of packaging material which are usually used for fragile products such as styrofoam, peanut foam, PE foam, bubble wrap, kraft paper, honeycomb, and biodegradable packing made of natural, non-toxic sources such as wheat, mushroom, and cornstarch. The common of these materials provide suitable cushioning that prevent damages in inside areas and outside areas due to hazards and collisions during transit from the manufacturer to customers or the retailer, or even incidents while the products are in the manufacturer warehouse or on retail shelves. Traditional packing with styrofoam, peanut foam, PE foam, or bubble wrap has many advantages such as low costs, lightweight, water insulation, easy to handle characteristics, and flexibility to return to the original shape after absorbing shock. However, they are types of plastic which create burdens in solving problems of long-term waste treatment, and are harmful for the environment and human health. While different types of kraft paper, honeycomb, and biodegradable materials can be the right solutions for these issues, they do have their disadvantages, i.e., higher weight and higher costs than traditional packing. Therefore, most companies are not ready to 100% replace their current package material to biodegradable materials. Instead of using full plastics-based protectors for packaging, they try to reduce the number of plastics and combine them with friendly environment material.

In Vietnam, current packaging methods for heavily fragile products are commonly styrofoam and carton box. The packaging method is simple with styrofoam protectors for the product corners, tops, and bottoms to prevent them from being shocked or vibrations during transportation and handling processes. However, during the storage and transferring products styrofoam protectors can be broken down into small pieces due to the pressure and collision between pallets. These are annoying for the customer when opening the box and have a lower rating for professional packing. Increasing quality and professionalism in product packaging is a current trend in management strategy, especially, when Vietnamese companies expand in exporting products to Europe and developed countries. However, the extra cost for packaging can result in significantly reduced profits of the company. The target of the study is to give a guideline on how to apply PDCA methodology to tackle problems in manufacturing. The objectives of the research focus on two main points. The first one is to develop a simplified combination of PDCA cycle and quality tools for quality packaging improvement. The second one is to deploy this PDCA cycle to solve multi-objective problems for packaging designs. The designs should guarantee protection quality, be friendly to the environment, and contribute to manufacturing cost reduction. Through the case study of finding the new packaging method which

uses eco-friendly materials to improve the quality, and reduce the defect ratio due to packaging, the authors prove that PDCA is an effective method for teamwork in order to solve problems.

## 2. Literature Review

Plan–Do–Check–Act (PDCA) methodology was first created and defined in the 1930s by the American statistical expert Walter A. Shewart [2,3]. Then, in the 1950s, it was developed by W. Edwards Deming and became one of the most well-known methods in the world to guide improvement [3,4]. The Check step is sometimes called Study, and the cycle becomes the Plan–Do–Study–Act, PDSA, cycle [5,6]. The "Plan" step includes analyzing and evaluating the existing situation. After identifying all possible and root causes, the area, opportunities for improvement are recognized and prioritized. Then, changes in the system and setting targets that relate to improvement are proposed and planned. In the "Do" step, the changes are carried out, usually on a small or pilot scale to obtain the results for studying and analyzing. For each change, the idea is tested and data is gathered to support the next phase to compare before and after effectiveness. The "Check/Study" step consists of analyzing the results of the changes, determining learning lessons from carried out changes, comparing with setting targets to see whether solutions brought adequate results. In the "Act" step, if changes lead to improvements, they are adopted and applied on a larger scale. Otherwise, they are abandoned. The process can be iterative and may require several cycles for solving complex problems. In general, the PDCA cycle is a continuous process shown in Figure 1, i.e., it is not an end-to-end process. When you reach the last stage of Act and the outcomes meet the planned targets, you should start all over again and constantly look for better and continuous improvements.

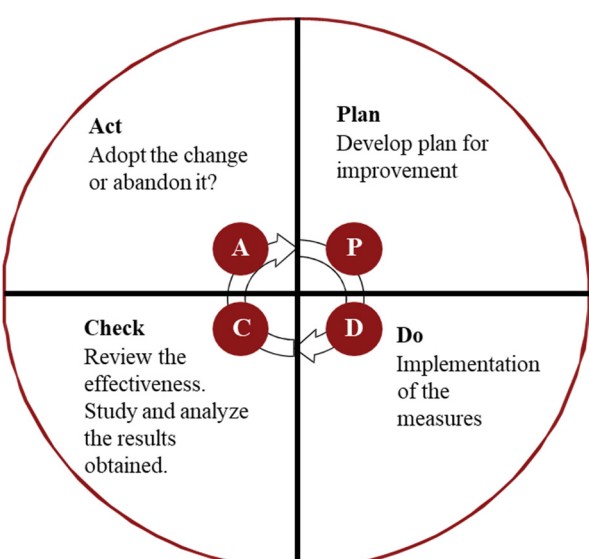

**Figure 1.** Plan–Do–Check–Act (PDCA) cycle.

Various quality tools can be used as support for an effective PDCA cycle [7]. They are 5 Whys, 5S, 5W1H, or 5W2H, brainstorming, Ishikawa diagrams, Pareto chart, the flow chart, Poka Yoke, Six Sigma, Failure Mode Analysis and Effects, Fault Tree Analysis, etc. More than a simple approach, PDCA is a philosophy that should be applied in the organizational culture for continuous improvement. For each case study, PDCA is a new intersection between the scientific method and specific problem-solving actions. Various scientific methods, quality tools are used for different phases of PDCA to reach the best possible improvement. The art of utilizing tools in the PDCA cycle is a scientific and practical combination that currently attracts numerous industry professionals as well as scientists. In this paper, the 5 Whys, Ishikawa diagrams, 5W2H, Computer Aided Design (CAD) and prototypes are used during performing PDCA cycle. By mastering these tools, the article proposes a simplified

way that emphasizes the combination of root-cause analysis, active and creative ways to generate ideas in the planning phase, design for sustainability in the doing phase, and testing methods and implementing evaluations during PDCA cycle. The case study of this research can contribute as a successful application of PDCA in the packaging problems which normally lacks guidance for using systematic methodologies for quality management.

In recent years, the effectiveness of the PDCA methodology for quality improvement has been underscored in manufacturing [8], services, the health care system [9–11], and other environments. Matsuo and Nakahara examined and proved the positive effects of PDCA and on-job-training on significant improvements in workplace learning [12]. Wahiduzzaman, et al. used PDCA and 5S to minimize the sewing defects for the knit T-Shirts product in an "Interstoff Apparels Limited", Bangladesh. They concluded that PDCA cycle is an excellent tool in continuous improvement planning which resulted in increasing profit and quality for the company [13]. Realyvásquez, et al. used the PDCA cycle with support tools like the Pareto charts and the flowchart in finding a solution for at least 20% defect reduction. The successful implementation results in decreasing by 65%, 79%, and 77% defects in three product models [14]. Sunadi Sunadi, et al. implemented statistical process control through the PDCA cycle followed with Ishikawa diagram, 5W1H method, and nominal group technique to improve beer cans packaging. They analyzed and found out the solutions for the process capability index meeting the specification. The results showed that PDCA is a useful and effective method for improving packaging quality [15]. The benefits of practicing PDCA in planning, testing, and developing changes for target purposes make it become a well-known technique in the quality management field [16–19]. In general, literature shows that the PDCA cycle is highlighted as not only an effective tool for the quality control of products and process development but also as a logic program for continuous improvements [20–23]. In addition, PDCA can be used as a base approach to integrating with other methods in Lean and six sigma methodologies [24–26]. Garza-Reyes et al., proposes an approach, based on the PDCA cycle, to systematically conduct Environmental-Value Stream Mapping (E-VSM) studies. They successfully implemented the PDCA-based approach to improve the green sustainability performance of operations [27]. Jones et al., utilized the PDCA cycle to implement Six Sigma. They introduce a framework which operationalizes Six Sigma implementation with quality management and the PDCA cycle to effectively achieve executive commitment [28]. However, learning and using PDCA cycles can be challenging and time-consuming if the practitioners do not fully understand the methodology. Unsuccessful PDCA applications are the result of many reasons such as: Poor studies on a current problem and its obstacle, erroneous data collection, wrong or improper use of quality tools, fail in defining root causes, insufficient analysis, process non-standardization, or no sharing learning experience before and after the PDCA implementation [29,30]. This paper explores all phases of the PDCA cycle for a full understanding. Through a case study in improving packaging quality, the paper provides a benchmark example on how to apply PDCA in planning, reducing defects, checking, and monitoring further implementation steps. These help practitioners to avoid mistakes or main barriers, and to find the right implementing way as well as success factors for the PDCA cycle.

In the next part, quantitative and qualitative methods of the research are explained in detail. Table 1 is a summary of the methods and techniques used in phases of the proposed PDCA cycle. The applications of these methods and techniques in packaging quality improvement will be further discussed in the next session.

**Table 1.** Research methodologies.

| PDCA | FlowStep | Stage | Methods and Techniques | Goal |
|------|----------|-------|------------------------|------|
| Plan | 1 | Problem definition and clarification | Interviews Questionnaires Analyze quality report | Clearly defining problems of current packaging methods Define customers' requirements |
| | 2 | Data collection | Check sheet Observation record | Investigating the current situation, obstacles, specific characteristics of the problem |
| | 3 | Analysis | Brainstorming 5 Whys technique Ishikawa diagram | Discovering all possible causes, the n figuring out root causes |
| | 4 | Action plan | Brainstorming 5W2H technique Gantt chart | Generating and evaluating potential countermeasures to eliminate root causes Conceiving detail plan with clearly time bounds to block root causes |
| Do | 5 | Material testing | Market research Cost comparison Implementing tests to check/compare durable ability, resist bursting, the strength of materials | Finding eco-friendly materials for packaging which meet the technical requirements and at the acceptable price |
| | 6 | Design | Brainstorming and affinity diagram Evaluation and selection techniques Computer aided design Prototype | New packaging design to eliminate/block root causes Apply solutions to real products |
| Check | 7 | Check | Brainstorming Drop tests Doing experiments | Checking if the countermeasures were effective? if not, return to the Do phase |
| | 8 | Record | Comparison and evaluation techniques Data collection | Analyzes and compares experiment results with the target goals |
| Act | 9 | Standardization and conclusion | Flow chart Standard operating procedure | Standardize the improvements Sharing learnings and achievements Plan for further steps |

## 3. Applying PDCA Cycle in Packaging for Quality Continuous Improvement

The research applies a PDCA methodology to improve the packaging quality for sample fountain products of a GPEM's industrial partner. The company produces and exports composite planters, fountains, and other gardening things. Based on its monthly reports, the primary defects caused by inadequate packaging methods are cracks on the bottom, rim, and body of round shape fountain products. These products have fine-stone surfaces and weight from 11 to 20 kg. In addition, the current packaging method also is complained about using non-environmentally friendly materials. Cushions are easily broken or displaced from the original position during transportation. Using PDCA methodology which is a simple and powerful tool in tackling the problems and improving the team problem-solving sessions, will help systematic organizing team members' thoughts and implementing actions. Although PDCA descriptions are fairly easy to understand, to effectively apply it, it requires the involvement of all team member's responsibility. In addition, it involves multiple layers from analysis to testing. Therefore, the members have to cooperate well in informative commutation and implementations. Through the case study, this paper will cover each step of PDCA as well as recommend you with tools for accelerations and increasing effectiveness.

*3.1. Plan*

The first step in the PDCA cycle is Plan. This step includes defining problems and collecting all relevant data. Then the team has to find out the problems' root causes to develop an actionable

tested plan. However, identifying the key stakeholders and understanding customers' expectations are mainly focused first of all. Since the PDCA requires many steps from defining, planning, testing, analyzing, etc., a multidisciplinary or cross-functional team should be established. This team includes members who have different main functions, who can communicate and interact with each other frequently toward the main goal. The Plan phase normally consumes more time than others due to it should be done very carefully in clarifying the problem, finding and analyzing the root causes, developing solutions or countermeasures in an action plan.

Problem clarification–root cause analysis: The crack report of a GPEM industry partner shows the increasing percentage of the cracked packages from the pre-shipment stage to customers' storage checking. Cracks appear on the rim, body, and also the bottom of products. These problems happen to different types of products. In general, the higher percent of failure ratio, the more cost penalty the manufacturer has to pay. Because of the damaged product, the company has to be responsible to customer for warranty or change the new product. Especially, from the first three months—the highest market demand duration this year, the cracks at the body, bottom, and top of the products is 1.64%, which is extremely high compared to previous periods. There may be a consequence of bad packaging methods or wrong manipulation that leads to ongoing problems. In addition, current consumers and industry trends which prefer using sustainable materials for packaging also put the company in the need of improving the packaging sustainability. Therefore, finding the materials and packaging methods which are friendly to the environment and still ensure bringing profit for the company are in urgent need. Table 2 is an example of how the team recorded and analyzed the actual situation to identify current problems in packaging.

In Plan step, the re are several useful tools to improve the efficiency in team communication and problem-solving.

**5 Whys technique**: The simplest tool for solving problems, developed by Sakichi Toyoda, a Japanese inventor and industrialist. The basis of this effective approach is to ask why five times for a specific problem. Taiichi Ohno, a famous quality guru, believes that "by repeating why five times, the nature of the problem as well as its solution becomes clear". However, in general, you may need to ask the question fewer or more times than five before the root cause of a problem is found. The meaning to strive for 5 whys is to not give up easily until finding a root cause instead of ending up with a "symptom". Beside be used individually, the 5 Why technique is usually as a part of the Ishikawa diagram to identify the causes.

**A cause and effect diagram**: This is also called a fishbone diagram or an Ishikawa diagram. Using this diagram for a given problem, all its possible failure causes can be identified, categorized, and displayed. In the diagram, the right or its head shows the problem or effect which is identified. There is a spine, drawn by straight lines and big bones or Ribs. These bones show the relationship between major causes and the effect. Team members will need brainstorming (or use the first Why) to define the major causes of the problem. Medium size bones show secondary causes, small bones represent root causes. Ishikawa diagram is used to evaluate the root causes and to brainstorm solutions to them.

**5W2H**, five W's two H's is a method for asking questions about a process or problem. 5W2H represents: Who, what, when , where, why, how, and how much (or many). It usually is used when defining or analyzing a process or a problem for improvement opportunities as well as planning. When applying 5W2H method, the problem or situation has to be reviewed. Then appropriate questions about the problem are developed. The order of 5 W's and 2 H's is not important. A question and answers can lead to additional questions to form the improvement plan or to generate possible changes.

**Table 2.** An analysis example of grasping the actual situation, identify current problems in packaging.

| | |
|---|---|
| 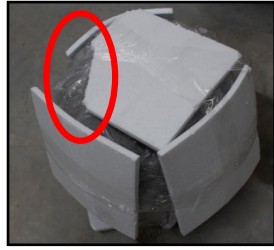 | For spherical shape product, the re are four foam boards which cover all sides and are stuck by tape. The boards are easily broken down.<br>Problem: Styrofoam is damaged during storage or delivery; The flat protectors are easily be shoved away from their initial positions; Using unfriendly environmental material |
| 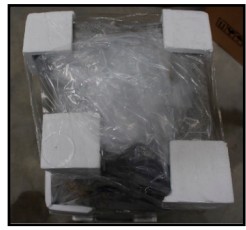 | For rectangular cuboid type, four L shape Styrofoam are on the top and bottom of the product, the other four foams are placed in the middle.<br>Problem: Styrofoam often is broken down in pieces. This does not only annoy customers but also reduce protection. |
| 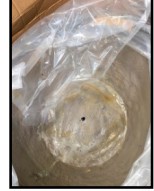 | Open one box for checking: Dirty inside nylon covered product |
| 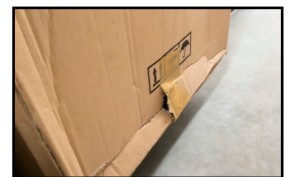 | Outside: Boxes are bent by overloading on the top |
| 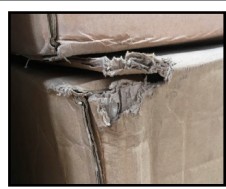<br>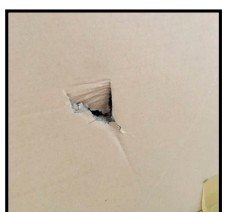 | Outside: The carton box is burst/torn or scratched by collisions. |

Figure 2 is a developed Ishikawa diagram for the issue: Cracks on the products. The team uses the first Why question to define the major causes, i.e., transportation, storage, and packaging. With each major category, additional whys questions are used until finding out the root cause. Normally, the finish goods after checking quality and packaging, the y are stored in inventory then finally delivered to customers. Cracks can happen due to incidents related to human or machine factors during the storing time or on the delivery way. Or the effects caused by poor packaging method. The real arrangement in the warehouse is not followed by regulation. In one pallet, the re are different kinds of products. They are not arranged with the same quantities and sizes. The overloaded container is one of the reasons to cause carton box failure. These arrangements are the result of insufficient training. Careless handlings in the warehouse when loading or unloading pallets are caused by

untrained employees. For the second major group cause which relates to transportation, the bad road conditions and truck driver steering habit are also possible reasons. The third major group related to the packaging method brings many possible causes of cracks on products. They are inadequate cushions, low-quality carton box, a poor fixing level between product edges and box, and ineffective separations. These potential factors are continuously dug deeper to find the root or initial causes by using 5 Why technique.

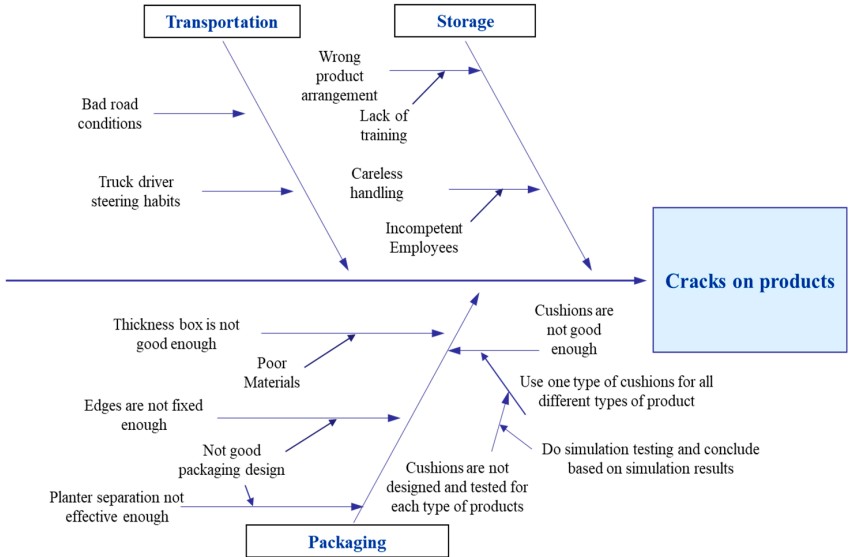

**Figure 2.** Ishikawa diagram for possible causes creating crack.

For example, cracks on the product:

1st W: Why are they happening?

Possible causes can be bad transportation, storage or packaging

2nd W: Why have bad packaging?

One reason can be that cushions in the packages are not good enough.

3rd W: Why using not suitable quality cushions?

Because the company uses one type of cushion for all different types of products. Cracks can happen on parts where the cushion is not provided enough or not thick enough, or the shape of cushions is not fit with the product.

4th W: Why use only one type for all products?

Because cushions are not designed and tested for each type of products. There is no instruction for using extra numbers of cushions for different sizes of shapes of products.

5th W: Why not design and test cushion for each type of product?

The design team did simulation testing to saving cost and concluded the current cushions can be used for a wide weight range of the products.

After evaluating current problems, finding root- causes, realizing inputs, and desired outputs, the team has to set the target goal to lead the plan in the right direction.

Below is the target in this case study.

**Target:** *Improving packaging or finding a new solution with environmentally friendly material to tackle the problem of damaged products while providing cost-effective packaging.*

As the analysis above, with some types of products, the cushions inside the packing box are not suitable or not good enough. The design team might not use appropriate or essential methods to test cushions and the packaging method such as the drop tests for impact points. This can lead to two problems: Not being able to conclude exactly the cushion efficiency for different types of products. The cushion design itself (thickness, material, shape, etc.) may protect good for one type of product but not for others. When finding out these root causes, the team has to generate a solution to solve them.

However, the team also needs to consider the cost of producing and testing a current or a new design. 5W2H can be used to clarify all aspects of the problems and concerns and take them into account when developing solutions.

Table 3 can be an example of how to make questions by using 5W2H in creating new cushion design, developing the real- testing method, and implementing it.

**Table 3.** 5W2H (five W's two H's) example.

| 5W2H | Question |
|---|---|
| Who? | Who design and test the packaging cushions? Who should be involved and responsible for it? Who are involved but should not be? |
| What? | What are the essential requirements with a new cushion design? What are the mandatory tests before using new designs? What are the effective methods for real-testing? |
| Why? | Why we have to redesign cushions and do multi real-test for new cushions and packaging? |
| When? | When the activities (redesign and practical testing) are started? |
| Where? | Where are these activities done? |
| How? | How are these done? |
| How (much, long)? | How much do they cost? How long do they take? |

When generating and evaluating the possible solutions, with the 5W2H tool, the team can consider their feasibility, practicality as well as economically. 5W2H is not only useful for planning, but it also brings benefits for other phases in the PDCA cycle. Table 4 is a PDCA timeline with detailed tasks for the case study. 5W2H can be used for finding viable solutions, planning necessary resources, designing experiments with expected outcomes as well as defining methods for measure performance

**Table 4.** PDCA timeline and action plan for the case study in packaging quality improvement.

| PDCA | Detailed Tasks | | | |
|---|---|---|---|---|
| Plan: 2 Sept–13 Sept | -Figure out customer requirements for packaging -Grasp the actual situation, identify current problems in packaging -Perform root causes analysis -Set and clarify target goals -Propose solution/countermeasures | -Output: An action plan for quality improvement and to use sustainable material in packaging which includes: -Plan to test the current packaging material and compare it with other types (one or more than one) which have a competitive price -Propose a new packaging material: Minimum use of environmentally unfriendly packaging material. -Timeline to create new designs for packaging -Plan for doing real tests for materials and completed packaging of products | | |
| Do: 16 Sept–20 Sept | | Implement actions | Carry out proposed improvement actions or countermeasures which include: -Finding new materials for packaging -Using CAD/CAM software to design protectors for planters and fountains -Implement solutions to real products | |
| Check: 23 Sept–27 Sept | | | Check the achievements | -Drop test to check the performance of new packaging -Verify the results |
| Act: Oct | | | | Make decision: Adopt or reject changes? | -Standardize the improvements -Document benefits -Define future Plans which include: + Propose company to apply solutions into real packaging line + Plan for continuous improvements |

*3.2. Do*

In the Do phase, the action plan in the previous stage is carried out. The team firstly finds new material or better quality material for packaging which is biodegradable and recyclable. Cardboard and honeycomb are viable materials to replace styrofoam. However, they may create more costs, especially with honeycomb material. After doing the market research for new viable material, price, and available suppliers, the design team decides to use corrugated cardboard which has three to five layers. The more layers a cardboard has, the more expensive it is. The five-layer corrugated fiberboard is used in packaging heavy fragile products.

Table 5 is an example of how the team compares corrugated cardboard and carton boxes from different suppliers. There are several tests to check the durable ability, resist bursting, the strength of carton boxes and cardboards. The results from the compression and bursting tests (Table 5) show that although having the same price, the quality of carton boxes is different from many suppliers. Ojitex carton box and cardboard are much better, more durable than the current ones. Therefore, the team decides to use Ojitex carton boxes and cardboards.

**Table 5.** Carton box testing.

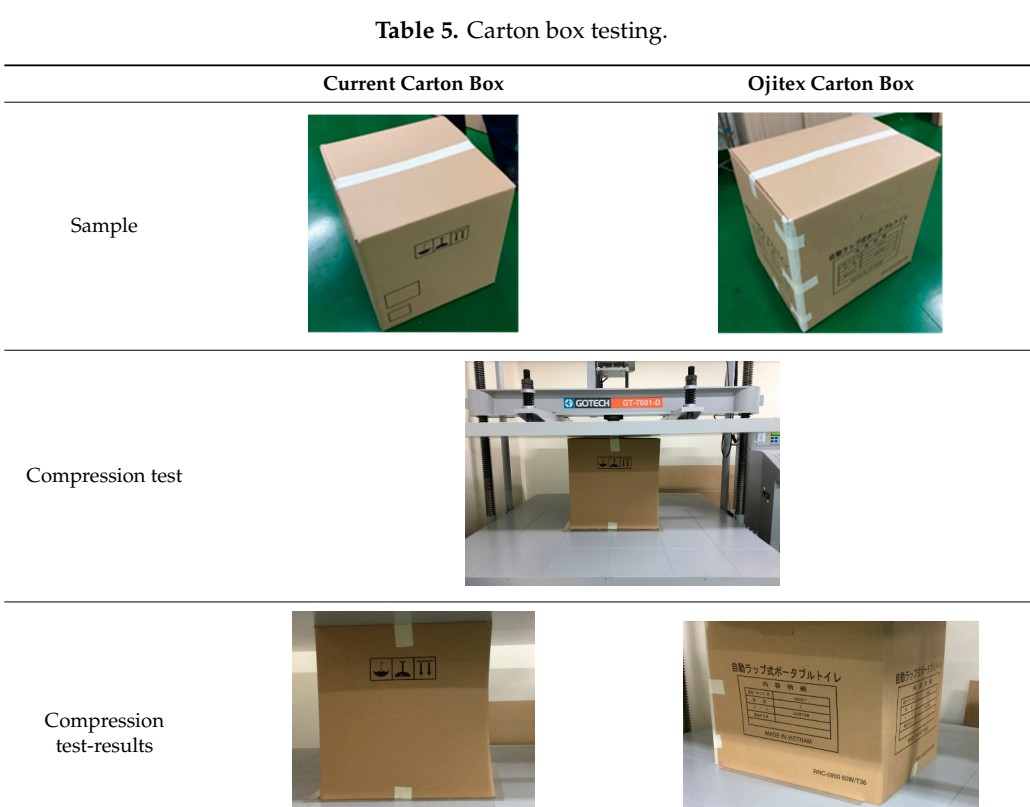

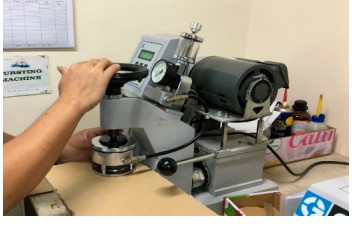

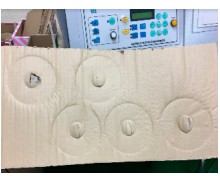

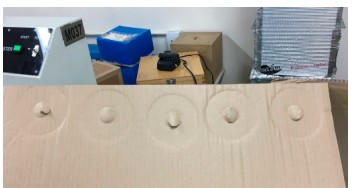

| | Current Carton Box | | Ojitex Carton Box | |
|---|---|---|---|---|
| Bursting test-results | | | | |
| | Time | Bursting force (kg·f/cm$^2$) | Time | Bursting force (kg·f/cm$^2$) |
| | 1 | 8.0285 | 1 | 12.6824 |
| | 2 | 7.6124 | 2 | 14.2746 |
| | 3 | 7.6741 | 3 | 13.6171 |
| | 4 | 8.2956 | 4 | 14.2695 |
| | 5 | 7.4789 | 5 | 13.1754 |
| | Average | 7.8179 | Average | 13.6038 |

Deform force (kg·f): 321.91      Deform force (kg·f): 753.98

Sample · Compression test · Compression test-results · Bursting test · Bursting test-results

Another important action in the Do phase of this case study is to change packaging design. The new design has to maximize the advantages of new material and help in cost reduction. Brainstorming and affinity diagram are two techniques recommended for teamwork to generate creative solutions. While brainstorming encourages team members to come up with ideas in a free and open environment, the affinity diagram helps to organize these data into groupings consolidated information. Using the advantage of science and technology in this stage helps shorten times in designing or implementing. In this case study, mesh carton structure designed to provide both cushion and linkage protection. This protection prevents damages from forces acting not only from sides but also from the top and bottom of the carton box. When people load boxes to pallets and store them in a long time, this packaging style help to balance pressure on the contact area to all points of bottom or top side. Figure 3 shows a 3D packaging design using computer-aided design software. Figure 4 is a real prototype of a new packaging design.

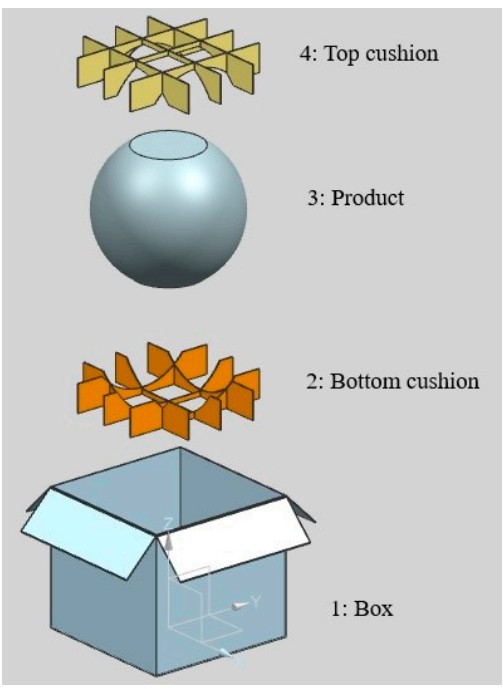

**Figure 3.** 3D Computer Aided Design (CAD) packaging design.

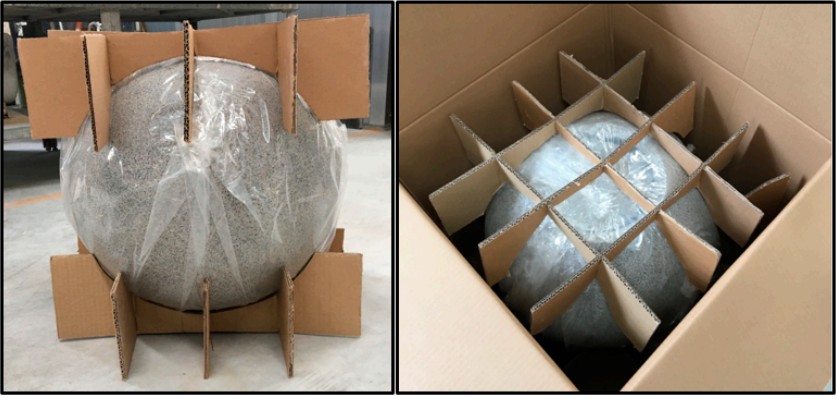

**Figure 4.** Real packaging.

*3.3. Check*

At the Check stage, the team analyzes and compares experiment results with the target goals stated in the plan step. The check step is essentially important in decision making and defining

further steps. For the case study, the package must resist drag, stretch, torn, and drop conditions by manufacturer's standard. Then, when it is delivered from the production site to the customer's hand, the product inside still remains its function and appearance.

In packaging protection aspect, the se factors are mandatory and considered as main requirements to an individual package:

- Size of the box or the bottle where the object is packed inside;
- How the box can resist forces impacting on all sides;
- How it absorbs vibration and shocking condition during transporting; and
- What is the maximum protection level when the package is damaged, and whether it still retains the original product.

Based on these factors, the team developed a drop method to test the efficiency of the new packaging design. In this method, the packing piece (delivery packaging or master carton) is filled and sealed for transport. It is then lifted to the designated height and held there before it is dropped.

The number of dropping cycles to be executed: One carton box passes through 7 cycles, according to the 7-impact points (Figure 5). Dropped height is the distance between the lowest point of the packing piece and the impact surface. The height may not deviate more than ±2% from the designated dropped height (Table 6). The evaluation takes place after the execution of all seven cycles.

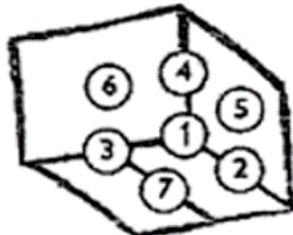

**Figure 5.** Sequence of the impact points in dropping test.

**Table 6.** Dropped height for a gross weight.

| Content | Dropped Height for A Gross Weight | | |
| --- | --- | --- | --- |
| | Up to Max 10 kg | Up to Max 15 kg | >15 kg |
| Fountain | 500 mm | 400 mm | 250 mm |

**Evaluating the new packaging method:** For mid-weight (under 15 kg) and round shape products: 100% of the packing boxes withstand the dropping test condition by absorbing shocking in both vertical and horizontal axis. This new packaging method fills the box's space, protects the inside product at weak points, and separates weight concentration. It protects the inside product when vibrating and shocking. However, with the product which weight more than 15 kg, there are still some small cracks on the top and bottom parts.

The cost for each packaging involves a carton box, plastics foil covering the product, and sub-items such as honeycomb board and adhesive tape. At present, the company spends 33,644 VND for each unit packaging. The new method requires 35,684 VND. When comparing with the current packaging method, the cost per unit of the new method increases by 6%. This number extremely satisfies the customer's requirement which states that the extra cost should not be higher than 20% compared with the cost of current packaging.

### 3.4. Act

In the Act phase, the team documents the results and makes the decision on adopting or refusing the changes. Be noted that PDCA is applied for continuous improvement, therefore, it is not a start-end process. At the Act phase, another plan to look for an even better-improved way should be continued.

In this case study, the new packaging design will be widely implemented for mid-weight (under 15 kg) and round shape products. A packaging standardization process is developed with detail task assignments (Figure 6). For the large weight (above 15 kg), the solution was not totally successful, the team rejected to applied it. Another PDCA cycle will start again to dive further into tackling the problem.

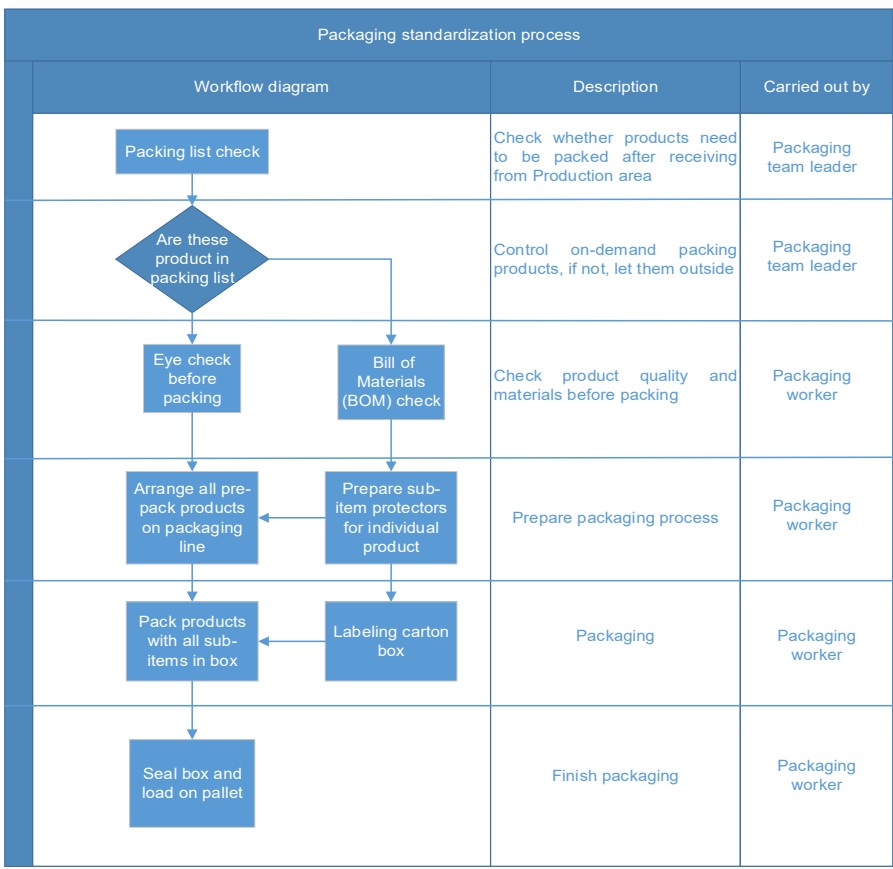

**Figure 6.** An example of a packaging standardization process.

## 4. Conclusions

This research highlights the benefits of PDCA methodology in quality improvement. It instructs the simplified way to practice this method with support tools such as the 5 Whys, Ishikawa diagrams, 5W2H, and Computer-Aided Design (CAD). Through the packaging case study, PDCA combined these tools and analyzed packaging defects, found root-causes, and facilitated development of more sustainable ways to tackle problems. The new packaging method not only uses environmentally friendly materials but also extremely reduces the defect ratio. With respect to the objectives of the case study, the new design uses 100% recyclable, biodegradable materials. It can minimize surface crack defects for mid-weight products or significantly reduce the defect ratio in large-weight products due to incidents in delivering and storing. The extra cost for the new packaging method is slightly higher than the old one, but the benefits of a higher competitiveness and customers' satisfaction can be exploited. For long-run evaluation, the new packaging design can help saving costs from decreasing the numbers of rejected defects or reworked products, reducing the packaging time and labor forces. These result for increasing customer satisfaction, the company's profit, quality as well as reputation. The PDCA cycle should be replicated for quality improvement, reducing defects and driving towards sustainability for packaging methods. Recommendations to successfully apply PDCA methodology are mentioned in the paper. In the Plan phase, keys for effective implementation are team establishment, teamwork spirit, using proper quality tools for gathering data, clearly defining problems, and the analysis of the current

situation as well as of obstacles. The outcome of the Plan stage should include root-cause analysis, potential solutions, and a detailed implementation schedule with time-bounds. In the Do phase, implementing, refining, and finalizing countermeasures are essential actions. Using the right tools and techniques for these tasks will result in shortening the testing times and saving costs. In the Check stage, proper methods and plans for evaluations help to avoid erroneous data collection which can lead to wrong decisions in the Act phase. Finally, the Act phase should include tasks in documenting results, process standardization, sharing learnings, and planning of further steps.

In conclusion, this research contributes understandable guidance with a successful benchmark application of PDCA for packaging problems. Through a packaging case study, it shows the art of combining scientific and practical methodology by utilizing tools in the PDCA cycle to achieve multipurpose objectives, i.e., responsibility in protecting the environment, increasing quality, and economic profit. Due to the limited number of studies about continuous improvements in the sustainable packaging field, the paper's application is expected to be highly useful for practitioners and researchers in similar quality projects. In addition, this simplified PDCA cycle with a combination of quality and design tools can effectively be applied in creating new designs, reducing defects as well as continuous quality improvement in any manufacturing field.

**Author Contributions:** Conceptualization, methodology, funding acquisition, supervision, V.N.; investigation, data curation, software, formal analysis V.N. and N.N.; project administration, visualization, writing—original draft preparation, T.T.; writing—review and editing, B.S. All authors have read and agreed to the published version of the manuscript.

**Funding:** This research was funded by The Vietnam Ministry of Education and Training, grant number B2019-VGU-02

**Conflicts of Interest:** The authors declare no conflict of interest.

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
