# Peer review of "Practical Application of Plan–Do–Check–Act Cycle for Quality Improvement of Sustainable Packaging: A Case Study"

_applsci, doi:10.3390/app10186332_

Round 1

Reviewer 1 Report

The authors discuss a practical application of the PDCA methodology.

The paper presents no research novelty. It is simply a success story in which they apply techniques such as : 5W2H, 5Why, Ishikawa diagrams, etc.

The problem is that hundreds of success cases can be performed and it is not of any scientific interest.

Reviewer 2 Report

The manuscript titled „Practical application of Plan-do-Check- Act cycle for quality improvement of sustainable packaging: a case study„ is a very interesting elaboration. The text deal with very important problem of the pakaging process improvement. The Authors have used the most popular  quality and improvement tools recommended by standards and international industrial organizations.

The manuscript has the proper form and structure. The title and abstract are enough informative. 
However, it is suggested to develop the analysis of the Ishikawa diagram - it should be more detailed - in this form it is not a sufficient source of the information. The Authors should highlight why these management tools were chosen and how the problems were identified (was VSM analysis performed for identification? As it is known, VSM it is the most commonly used tool of improvement activating PDCA.

Due to the universality of the subject matter of the manuscript, it can be assumed that the literature review also needs to be developed and improved (quite a limited number of citations and the lack of the highest world positions).

Despite pointed remarks, I have believe that the manuscript is a good source of information and after mini-revision I recommend it for publication.

Reviewer 3 Report

The present article addresses the problems of packaging heavy fragile products in the form of a case study.

The research applies a PDCA methodology to improve the packaging quality for sample fountain products.

The proposed method of packaging is already known in the world.

The article is almost a methodological guide to practice and does not have the attributes of a scientific article.

The present article lacks the scientific benefits that need to be added to the article.

Round 2

Reviewer 1 Report

The authors have made changes to the original document and these changes have improved the quality of the document.

The case study is well described and analyzed but does not have a scientific article organization. I think it should be improved by following the structure:

Abstract: Abstract must contain results for describes the main findings (with important numerical values, if this is possible.) and conclusions (including the main results and advances the field of study).

INTRODUCTION. This section must answer the question: what is the problem? The article must be related to the scientific context and must include the objective and the hypotheses raised. You must unify sections 1 and 2.

MATERIALS AND METHODS.
The phases that have been developed to solve the problem are included. It is explained how the study has been designed and the techniques developed and how the data have been analyzed are incorporated. The reading of the paper is too chaotic without this structure. Please, you should include this section.

Section 3, includes methodology and results. It is not possible for both sections to be linked.

DISCUSSION: Results obtained are interpreted and related to the findings that existed before developing the study. In this section, there is a subjective judgment ("These data show that...", "From this it can be seen that...", etc.).This section should be included

CONCLUSIONS: This is a final assessment of the research. This section answers the question: What do the findings mean? I do not see this precept in the conclusions of the paper

Authors should substantially improve the structure of the paper. At this time, the paper is too chaotic and underestimates the quality of the paper.

Reviewer 3 Report

The present article lacks a summary of the scientific benefits of the proposed method of packaging products.

Round 3

Reviewer 1 Report

Dear authors, thank you very much for following my recommendations to improve the document.

I think that it has now evolved in the right way. I think it can be accepted for publication, except for:

  • Figures 1, 5 and 6 are not cited in the text.
  • You must explicitly cite Table 1 in the text.
  • Table 2 is not cited in the text.

Reviewer 3 Report

I no longer have any comments on this article.
